# The Extreme Heat Wave of Summer 2021 in Athens (Greece): Cumulative Heat and Exposure to Heat Stress

Dimitra Founda [1,*], George Katavoutas [1], Fragiskos Pierros [1] and Nikolaos Mihalopoulos [1,2]

1 Institute for Environmental Research and Sustainable Development, National Observatory of Athens, GR-15236 Athens, Greece; gkatavoutas@noa.gr (G.K.); fpierros@noa.gr (F.P.); nmihalo@noa.gr (N.M.)
2 Department of Chemistry, University of Crete, GR-71003 Heraklion, Greece
* Correspondence: founda@noa.gr

**Abstract:** The Mediterranean has been identified as a 'climate change hot spot', already experiencing faster warming rates than the global average, along with an increased occurrence of heat waves (HWs), prolonged droughts, and forest fires. During summer 2021, the Mediterranean faced prolonged and severe HWs, triggering hundreds of wildfires across the region. Greece, in particular, was hit by one of the most intense HWs in its modern history, with national all-time record temperatures being observed from 28 July to 6 August 2021. The HW was associated with extreme wildfires in many parts of the country, with catastrophic environmental and societal consequences. The study accentuated the rarity and special characteristics of this HW (HW2021) through the analysis of the historical climate record of the National Observatory of Athens (NOA) on a centennial time scale and comparison with previous HWs. The findings showed that HW2021 was ranked first in terms of persistence (with a total duration of 10 days) and highest observed nighttime temperatures, as well as 'cumulative heat', accounting for both the duration and intensity of the event. Exceptionally hot conditions during nighttime were intensified by the urban heat island effect in the city of Athens. Human exposure to heat-related stress during the event was further assessed by the use of bioclimatic indices such as the Universal Thermal Climate Index (UTCI). The study points to the interconnected climate risks in the area and especially to the increased exposure of urban populations to conditions of heat stress, due to the additive urban effect.

**Keywords:** heat wave 2021; Greece; air temperature; cumulative heat; UTCI; thermal stress

## 1. Introduction

Heat waves (HWs), namely periods of unusually hot weather, have been classified among the most dangerous weather phenomena globally, associated with profound adverse impacts on humans and natural systems. In some parts of the world, HWs are responsible for more fatalities than any other natural hazard [1]. Studies based on observations or models suggest that, in the warming world, such events are becoming more frequent, stronger, and last longer [2–7], while changes in the seasonal timing and geographical locations of HWs are also reported worldwide [8–10]. In the early 2000s, research studies by Easterling et al. [11] and Meehl and Tebaldi [12] reported on the projected increases in the occurrence, intensity, and duration of extreme heat events and HWs in the 21st century. The years that followed verified these projections, with a sequence of severe and record-breaking HWs hitting many parts of the world [13]. After the deadly HW of 2003 [14], Europe has witnessed a series of severe HWs, such as the HWs of 2007 in the eastern Mediterranean and Balkans [15,16], 2010 in Russia [17], 2017 in southern Europe and Iberia [18,19], 2019 in central Europe [20,21], and 2020 in Siberia [22], with devastating impacts on human health, the environment, agriculture, and local economies. During these HWs, national previous long-standing temperature records were exceeded in many countries. Unprecedented high temperature extremes are reported even in the Arctic Circle, as, for instance, a temperature

of 38 °C was registered at the Siberian Arctic town of Verhojansk (67.55° N and 133.38° E) in June 2020 [22]. Moreover, in a recent study, Fischer et al. [6] reported an increased probability of 'record-shattering' extremes in the coming decades, dependent on the future warming rate rather than the global warming level.

While recent extreme heat events and HWs may constitute an unprecedented experience in northern European countries, the countries of southern Europe and the Mediterranean area are more acclimatized to hot weather [23]. However, the Mediterranean belongs to those areas of the globe characterized as 'climate change hot spots' in regards to the observed and projected thermal risk as well as projected drought conditions [3,24–29]. Warming in the area is now higher than the global average by 20%, although the Mediterranean warming and global warming rates were similar until the 1980s [29,30]. Lionello and Scarascia [29] reported that according to model simulations, warming in the Mediterranean will be approximately 50% (and locally up to 100%) higher than the global warming rate in the summer. Likewise, Zittis et al. [31] found that the eastern Mediterranean was warming two times faster compared to the world average during the period between 1981–2019. The increased warming rates have resulted in an increased occurrence of extreme heat events in the area. Specifically, the analysis of HWs in 26 regions of the world since 1950 revealed that the Mediterranean ranks among the top 5 regions that demonstrate significant increasing trends in all key parameters of HWs, such as intensity, maximum duration, frequency, and cumulative heat [7]. Future projections based on Euro-CORDEX simulations indicate a large increase in intensity and length of Mediterranean heat waves by the end of the 21st century, according to all global or regional models and all emission scenarios [5,28]. Moreover, big cities of the eastern Mediterranean, such as Athens or Nicosia, belong to the top-ranked European capitals with respect to HW severity and urban thermal risk expected towards the end of the century [27,32].

Summer 2021 has been recognized as one of the most extreme summers globally, in terms of the occurrence of unprecedented, intense HWs, causing numerous unexpected deaths and ecological disasters [33]. The HW that hit the Pacific Northwest of the United States in late June 2021 established new high temperature records, astonishing the scientific community, with temperatures up to 49 °C in inland areas and 6 °C beyond previous records [34,35]. Strong anomalies in large-scale atmospheric circulation, comprising a preceding instability of the polar vortex, the formation of a tropospheric blocking ridge, and finally, adiabatic heating from subsidence, are reported to have triggered and intensified the episode [35].

Another severe HW developed over the Mediterranean in the late July 2021, associated with the transfer of hot air masses from Africa, prevalence of strong anticyclonic conditions at different levels, and sinking air masses, inducing compressional heating. The HW was particularly strong and persistent across southern Italy, Greece, and western Turkey. A new European all-time highest temperature record equal to 48.8 °C was reported in Sicily on 11 August 2021 [36]. Prolonged extreme heat, combined with drought conditions, favored the development of extended wildfires, destroying forests and properties, and forcing residents to evacuate. Greece, in particular, experienced three separate HWs during summer 2021. The heat wave that hit the country from the end of July through early August 2021 (HW2021 hereafter) was characterized as one of the most extreme HWs that Greece had ever witnessed, comparable with the historic HWs of 1987 [37] and 2007 [15] engraved in population memory, mainly due to their fatal impacts on humans, forests, and agricultural production.

The study analyzed key parameters of previously reported HWs, such as intensity, duration, and cumulative heat, and evaluated the severity of HW2021 on a centennial time scale using the historical climate series of the National Observatory of Athens (Greece). HW2021 was rated in relevance to all previous HWs at NOA from 1900 to 2021 and specifically to past historic HWs in the area. Moreover, the study assessed the exposure of humans to heat-related stress during the identified HWs, using the advanced bioclimatic index UTCI (Universal Thermal Climate Index).

## 2. Materials and Methods

### 2.1. Study Area and Data

To reach our aims reported above, we used the long-term climate series derived from the historical records of the National Observatory of Athens (NOA), spanning more than one century. These are the longest climate records in Greece and some of the longest in the eastern Mediterranean. NOA is located on the Hill of Nymphs (37°58′ N, 23°43′ E, 107 m a.s.l) in the center of the city of Athens, near the Acropolis. The climate station has been operating uninterruptedly at its present location since the late 19th century. Details on NOA's historical climate station and records are included elsewhere [38].

Time series of daily maximum, daily minimum, and daily mean air temperatures ($T_{max}$, $T_{min}$, and $T_{mean}$) over the period between 1900–2021 were used for the calculation of climate indices and HW metrics. Additionally, hourly time series of air temperature, relative humidity, wind speed, and global solar radiation at NOA were used for the estimation of hourly values of the thermal index, UTCI. The analysis of the UTCI was restricted to the period between 1960–2021, following the availability of continuous measurements of solar radiation at NOA [39].

Monthly time series of daily minimum temperatures for the period between 1976–2021 at the coastal urban station of Helliniko (HEL), the suburban station of Tatoi (TAT), located in the northern outskirts of Athens, and the rural station of Tanagra (TAN), located to the north of Athens and at a distance of approximately 40 km, were also used in the study to account and discuss urbanization effects at NOA in relevance to the observed, prominent warming rates since the mid-1970s. The stations belong to the network of the Hellenic National Meteorological Service (HNMS).

### 2.2. Heat Wave Indices

Different criteria appear in the literature to identify a heat wave, widely perceived as a prolonged period of extreme heat [40]. The range of different approaches often causes the study of HWs across different sectors or areas to be complex [40]. The present study adopted a simple definition to identify a heat wave, assuming a minimum duration of three consecutive days to identify a 'prolonged' event and a $T_{max}$ threshold at or above the fixed 95th percentile of the summer (June–August) daily $T_{max}$ distribution at NOA for the reference period between 1971–2000, to designate 'extreme heat' [40–42]. This threshold value corresponds to 37 °C at the NOA station.

A number of metrics were chosen to characterize each HW identified by the adopted definition over the period between 1900–2021, in terms of the intensity, duration, and cumulative heat across the event. The highest temperature (hottest day) observed during the event represents the 'amplitude' of a HW (HWA), and the average temperature across all HW days represents the HW intensity (HWI) [7]. The amplitude and intensity of the HWs were estimated for all three air temperature indices, $T_{max}$, $T_{min}$, and $T_{mean}$ (estimated by averaging 24 hourly values of air temperature) as follows:

$$HWA_{max} = \max(T_{max,i}), \ i = 1, n \tag{1}$$

$$HWA_{min} = \max(T_{min,i}), \ i = 1, n \tag{2}$$

$$HWA_{mean} = \max(T_{mean,i}), i = 1, n \tag{3}$$

$$HWI_{max} = \sum_{i-1}^{n} T_{max,i}/n \tag{4}$$

$$HWI_{min} = \sum_{i-1}^{n} T_{min,i}/n \tag{5}$$

$$HWI_{mean} = \sum_{i-1}^{n} T_{mean,i}/n \tag{6}$$

with n representing the number of consecutive days comprising each HW event, namely the HW duration (HWD).

While HWs are often assessed through their impacts, from a climatic perspective, both intensity and duration mainly determine the severity of a heat wave. In this respect, rating a shorter and stronger HW against a longer but weaker HW is complex. Cumulative heat (Heat$_{cum}$) is an additional index that combines the intensity and duration of a HW [7,43–45] and actually constitutes a measure of the 'excess' heat during the whole event. It is estimated as the sum of the differences of daily temperatures from the predefined temperature threshold across all HW days. Heat$_{cum}$ was estimated for the daily maximum, minimum, and mean temperatures for each HW event, according to Equations (7)–(9).

$$\text{Heat}_{cum,max} = \sum_{i=1}^{n} \Delta(T_{max,i} - T_{max,P95}) \tag{7}$$

$$\text{Heat}_{cum,min} = \sum_{i=1}^{n} \Delta(T_{min,i} - T_{min,P95}) \tag{8}$$

$$\text{Heat}_{cum,mean} = \sum_{i=1}^{n} \Delta(T_{mean,i} - T_{mean,P95}) \tag{9}$$

with $T_{max,P95}$, $T_{min,P95}$, and $T_{mean,P95}$ representing the fixed 95th percentiles of the summer daily $T_{max}$, $T_{min}$, and $T_{mean}$ distributions for the reference period, corresponding to 37.0 °C, 25.7 °C, and 30.4 °C, respectively.

The consideration of $T_{min}$ and $T_{mean}$ (in addition to $T_{max}$) in the aforementioned HW metrics aimed to assess and highlight the excess heat levels through the nighttime and the whole 24 h period during the HWs. Exposure to elevated nighttime temperatures has been found to significantly increase heat-related mortality [46].

### 2.3. Thermal Stress Indices

Bioclimatic (or thermal) indices provide a measurement of human thermal discomfort induced by the thermal environment. In this study, the UTCI was employed to assess heat stress during HW events. The UTCI is based on a human heat balance model [47,48] and actually represents the 'physiological' heat stress of the human body in the effort to keep a thermal equilibrium with the surrounding environment [48,49]. The scale of the UTCI (expressed as the equivalent temperature in °C) in relevance to heat-related stress categories is shown in Table 1.

**Table 1.** The scale of heat-related stress according to UTCI. Source [49].

| UTCI (°C) | Stress Category |
| --- | --- |
| >46 | Extreme heat stress |
| 38–46 | Very strong heat stress |
| 32–38 | Strong heat stress |
| 26–32 | Moderate heat stress |
| 9–26 | No thermal stress |

In particular, the UTCI is based on a multi-node thermo-physiological model [47] coupled with an adaptive clothing model [50]. The index has been recommended to successfully assess the variability of thermal comfort/discomfort across Europe, along with the associated health impacts [51,52]. In addition to air temperature, the UTCI uses extra meteorological and atmospheric parameters that determine the human's perception of thermal stress, such as the air humidity, solar radiation, and wind speed. Hourly values of the UTCI during HWs were estimated from the hourly values of the aforementioned atmospheric variables, along with the mean radiant temperature at NOA. The calculations of the mean radiant temperature were performed with the RayMan model [53,54], employing the preceding meteorological parameters along with the longitude, latitude, and

elevation at the NOA station, as well as the time zone, local time, and day of the year. The analysis of the UTCI was restricted to the post-1960 period, on account of the availability of solar radiation measurements at NOA. A measure of the accumulated heat stress during HWs was assessed from the total number of hours under heat stress (at least strong/very strong/extreme), according to the scale in Table 1, across each HW. Finally, the maximum number of consecutive hours under strong/very strong/extreme heat stress during each episode was estimated and compared between HWs.

## 3. Results

### 3.1. Evolution of Summer Air Temperature and Heat Waves at NOA (1900–2021)

Figure 1a displays an updated graph of the summer (June to August) $T_{max}$, $T_{mean}$, and $T_{min}$ at NOA from 1900 to 2021. The temporal variability of all three air temperature indices depicted a wavy pattern marked by a succession of colder and warmer periods, for instance, the cold period at the beginning of the 20th century, followed by a warmer period from about the 1930s to 1960s, then a cooling period prevailing until the mid-1970s. Then, the summer air temperature at NOA exhibited a prominent and statistically significant increasing trend ($p < 0.001$) until the present, at rates of +0.56 °C/decade ($T_{max}$), +0.66 °C/decade ($T_{mean}$), and +0.77 °C/decade ($T_{min}$) over the period between 1976–2021. $T_{max}$ and $T_{min}$ increased asymmetrically, with higher warming rates corresponding to nighttime ($T_{min}$) temperatures. While this is a global phenomenon observed in most regions of the planet [55], the increasing $T_{min}$ at NOA has been also associated with increasing trends in the intensity of the nocturnal urban heat island (UHI) effect in the city [56]. In order to provide more insight on this, we further examined the evolution and trends in summer $T_{min}$ at additional stations of different characteristics in the area (as described in Section 2.1) over the last warming subperiod extending to the present (1976–2021) (Figure 1b). As shown in the figure, the summer $T_{min}$ at the urban (NOA) and coastal urban (HEL) stations steadily grew higher by several degrees compared to the suburban (TAT) and rural (TAN) stations. Nevertheless, all stations in Figure 1b present statistically significant ($p < 0.001$) increasing trends in summer $T_{min}$, at rates comparable to the warming rates at NOA, suggesting the regional character and spatial extent of the prominent recent warming since the mid-1970s.

The application of the definition of a HW, as described in Section 2.2, identified 67 individual HW events during the period between 1900–2021 at NOA. Figure 1c presents the frequency (number) of individual HW events per year over the study period, along with the mean duration (in days) of all HWs across each year and the maximum duration, corresponding to the longest HW in each year. As shown in Figure 1c, HWs were observed only in 44 out of the total number of years (122) of the study period. One single HW event was detected in 30 out of the 44 years, while two HWs/year and three HWs/year were detected in eight and five cases, respectively. The highest number of HWs (five in total) was recorded during summer 2012, which also represents the hottest summer on record until the present [41]. We also found that the vast majority of HWs (67%) lasted for three or four days, 21% lasted for five or six days, and only eight HWs (12%) lasted for at least seven days, with six of them occurring during the last two decades.

The long-term changes in the occurrence of HWs are further highlighted in Figure 1d, presenting the total number of individual HW events, along with the total number of days participating in HW events (HW days) during four consecutive thirty-year-long climatic subperiods, 1901–1930, 1931–1960, 1961–1990, and 1991–2020. Note that the years 1900 and 2021 were not included, to ensure time intervals of equal length. Cooler and warmer periods of the record as shown in Figure 1a correspond to lower and higher HW frequency, respectively. Low HW frequency was observed during the colder periods of the record (1901–1930 and 1961–1990), but higher in the warmer period, 1931–1960. A striking increase in the number of HWs and HW days was observed during 1991–2020, which was about five times higher than the average of the past three subperiods.

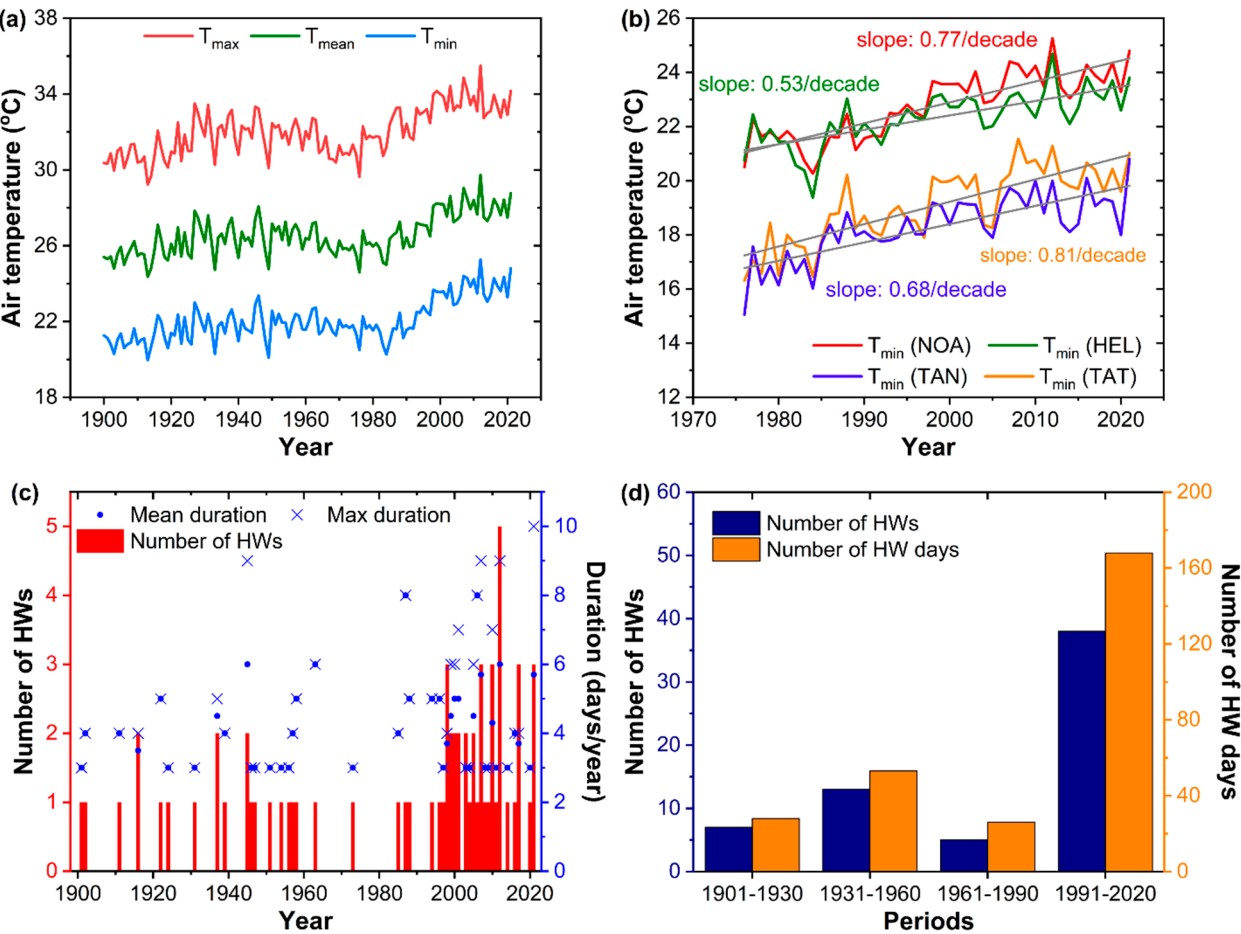

**Figure 1.** (**a**) The variation of summer $T_{max}$, $T_{mean}$, and $T_{min}$ in Athens (NOA) from 1900 to 2021; (**b**) the variation of summer $T_{min}$ at the stations of NOA (urban), HEL (coastal urban), TAT (suburban), and TAN (rural) from 1976 to 2021; (**c**) the number of HWs per year at NOA (1900–2021), along with the mean duration of HWs per year and the maximum duration (longest HW) per year; (**d**) the number of identified HWs at NOA, along with the total number of HW days, for four consecutive thirty-year-long climatic periods (1901–1930, 1931–1960, 1961–1990, and 1991–2020).

### 3.2. Heat Waves of Summer 2021

Greece was affected by a strong and persistent HW from the end of July to early August 2021. The episode affected the whole country, as shown in the map of Figure 2, presenting the highest $T_{max}$ values during HW2021 at a number of Greek stations spread across the country, derived from the networks of the Hellenic National Meteorological Service (HNMS) [57] and the automatic weather stations network of the National Observatory of Athens (NOANN) [58,59]. According to HNMS reports, previous long-lasting high temperature records were exceeded by up to 3 °C during HW2021 at nine continental stations of its network, operating since the mid-1950s [57]. Due to its large spatial extent, HW2021 affected both southernmost sites of the country (e.g., Moires and Sivas on Crete Island) and northernmost sites (e.g., Serres), as well as the islands of the Ionian Sea (western Greece) and Aegean Sea (eastern Greece), with temperatures exceeding 41 °C (e.g., Zakinthos, Samos). The $T_{max}$ at continental stations (e.g., Stylis, Makrakomi, Arfara, Sparti, and Argos) peaked above 45 °C, triggered by local geographical features, as well (local circulations, katabatic winds, etc.).

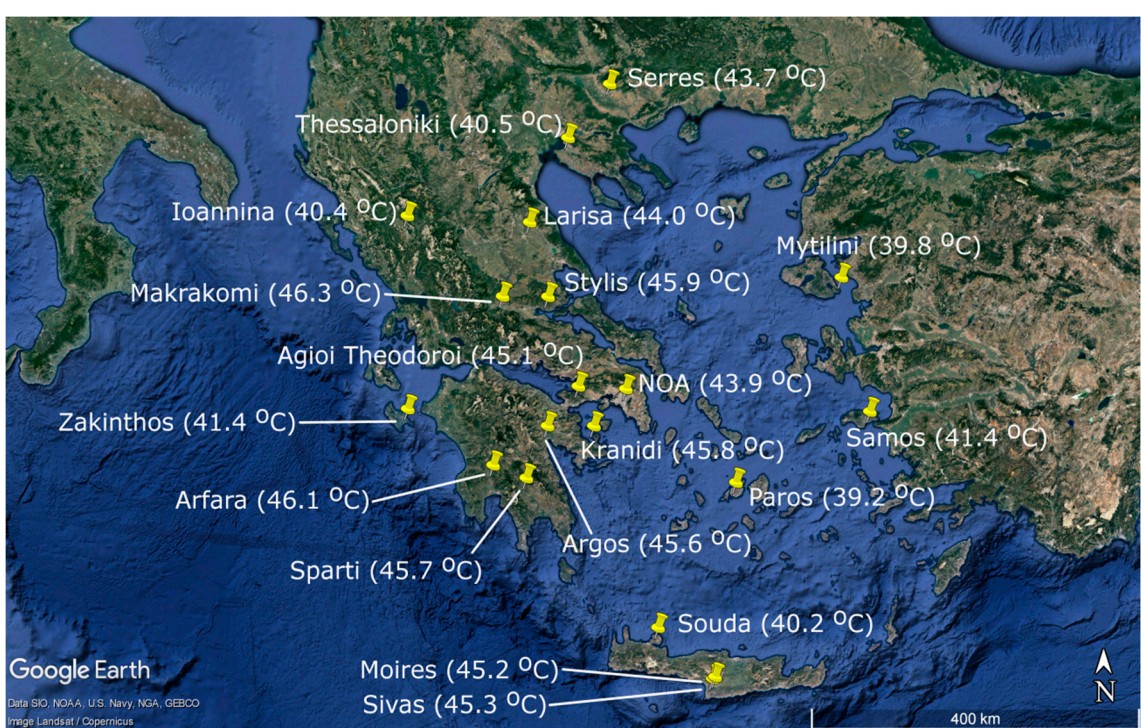

**Figure 2.** Highest $T_{max}$ values at several Greek stations during HW2021, according to the networks of NOANN (Arfara, Argos, Sparti, Moires, Sivas, Kranidi, Agioi Theodoroi, Makrakomi, and Stylis) and HNMS (Mytilini, Samos, Paros, Souda, Zakinthos, Ioannina, Larissa, Serres, and Thessaloniki), as well as NOA station. (The map was produced by Google Earth Pro).

Table 2 shows the top highest $T_{max}$ values (>45 °C) observed in Greece during HW2021, according to the NOANN network [58,59]. In almost all cases, the highest temperatures were observed on August 3. The table also includes the daily minimum temperature, along with the diurnal temperature range (DTR), namely the difference between the daily $T_{max}$ and $T_{min}$ for the same stations. The highest $T_{max}$ value was recorded at Makrakomi (46.3 °C). The corresponding values for Athens (NOA) are also included in Table 2.

**Table 2.** Top $T_{max}$ values above 45 °C during HW2021 in Greece, as recorded by NOANN [58,59], along with values at NOA station (bold fonts). $T_{min}$ and DTR values are also shown.

| Station Name | Latitude (° N)/Longitude (° E)/ Height a.s.l. (m) | $T_{max}$ (Date) (°C) | $T_{min}$ (Date) (°C) | DTR (°C) |
|---|---|---|---|---|
| Makrakomi | 38.9/22.1/125 | 46.3 (2/8) | 21.6 (2/8) | 24.7 |
| Arfara | 37.2/22.0/96 | 46.1 (3/8) | 27.8 (3/8) | 18.3 |
| Stylis | 38.9/22.7/55 | 45.9 (3/8) | 23.7 (3/8) | 22.2 |
| Kranidi | 37.4/23.1/110 | 45.8 (3/8) | 29.6 (3/8) | 16.2 |
| Sparti | 37.1/22.4/204 | 45.7 (3/8) | 24.4 (3/8) | 21.3 |
| Argos | 37.6/22.7/38 | 45.6 (3/8) | 25.3 (3/8) | 20.3 |
| Sivas | 35.01/24.8/96 | 45.3 (3/8) | 30.7 (3/8) | 14.6 |
| Moires | 35.03/24.83/54 | 45.2 (3/8) | 27.7 (3/8) | 17.5 |
| Agioi Theodoroi | 38.0/23.1/37 | 45.1 (3/8) | 27.3 (3/8) | 17.8 |
| **NOA Athens** | **37.9/23.7/107** | **43.9 (3/8)** | **31.6 (3/8)** | **12.3** |

Focusing on Athens and the NOA station, Figure 3a,b show the evolution of the daily $T_{max}$ and $T_{min}$ temperatures at NOA from 1 June to 31 August 2021 (92 days in total), along with the daily anomalies of $T_{max}$ and $T_{min}$ from their daily climatic means, based on the reference period between 1971–2000. According to the predefined criteria to identify a HW, three separate HW events occurred during summer 2021. A rather early, short HW with an

amplitude of 38.5 °C ($T_{max}$) was recorded from 26 June to 28 June, followed by a second one in mid-July (13–16 July), with the $T_{max}$ peaking up to 39.4 °C. A third, strong and persistent HW hit the country from 28 July to 6 August, with the daily maximum temperature at NOA approximating 44 °C on August 3 (Figure 3a) and the daily minimum temperature peaking up to 31.6 °C on the same day (Figure 3b). The $T_{max}$ at the NOA station was above 40 °C for 5 days during the episode. Anomalies of $T_{max}$ and $T_{min}$ exceeded 10 °C and 8 °C, respectively, across this last HW (HW2021) (Figure 3a,b). With the exception of the first half of June, the $T_{min}$ was above the mean climatic values by at least 2 °C during the whole summer period. Note that summer 2021 was the fifth all-time hottest summer at NOA in terms of $T_{max}$, but the second hottest in terms of $T_{min}$ temperatures.

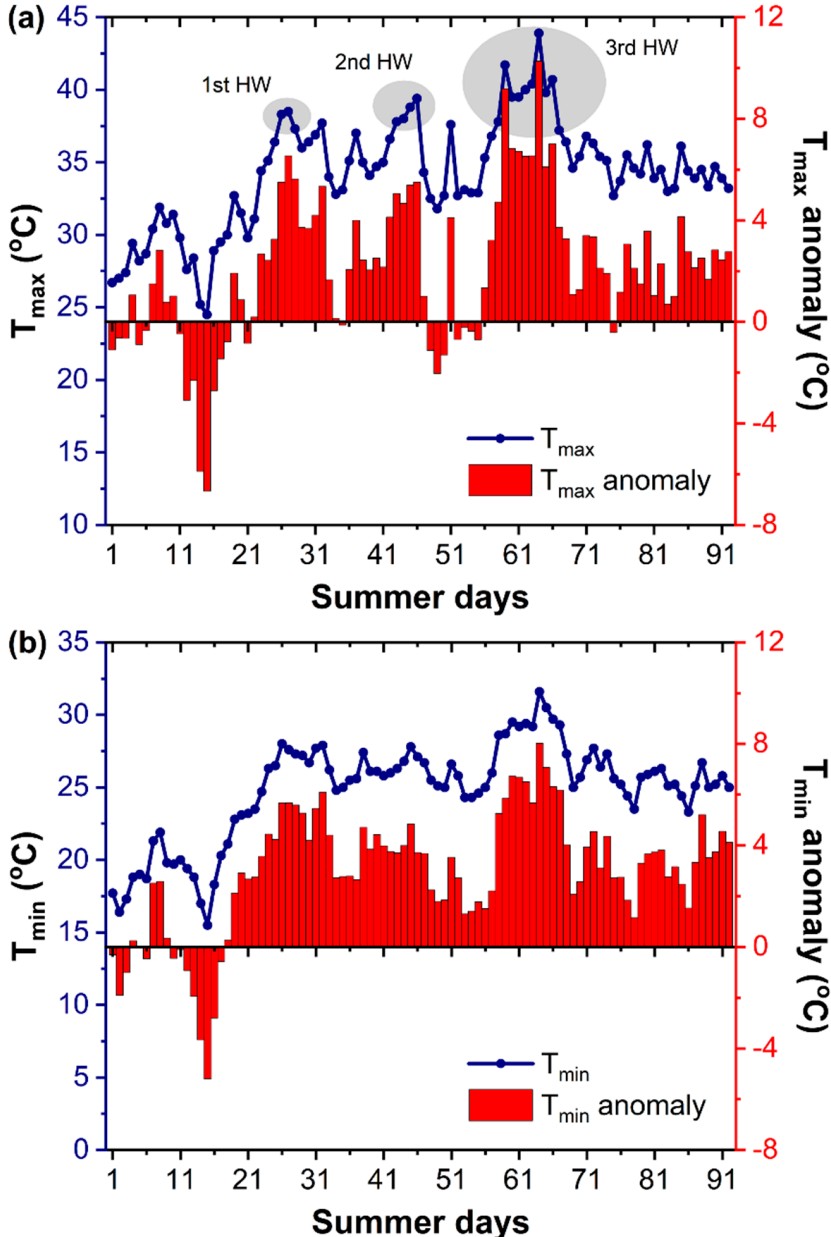

**Figure 3.** Variability of $T_{max}$ (**a**) and $T_{min}$ (**b**) at NOA during summer 2021 (1 June to 31 August), along with anomalies from the daily climatic means based on 1971–2000 reference period. The three HWs during summer 2021 are also marked on the graph.

The highest daily maximum temperature ($T_{max}$) during HW2021 at the urban station of NOA was lower compared to other Greek stations, as shown in Figure 2 and Table 2, in contrast to the nighttime temperatures ($T_{min}$), being much higher at NOA compared to other continental (or less urbanized) sites in the country. This is further illustrated in the values of DTR in Table 2, which ranged between 24.7 °C in Makrakomi to just 12.3 °C at NOA. The DTR ranged between approximately 14 and 24 °C across the other stations, determined by local effects.

The urban effect on the nocturnal UHI intensity at NOA during HW2021 was further quantified from the comparison of nighttime temperatures ($T_{min}$) between NOA and the rural station of Tanagra (TAN), as described in Section 2.1. According to Founda et al. [56], the magnitude of the nocturnal UHI, based on NOA and Tanagra measurements over a 35-year-long period (1970–2004), ranged between 4 °C and 5 °C in summer (with higher values of $T_{min}$ corresponding to NOA). This is further illustrated in the updated graph in Figure 1b, showing the summer $T_{min}$ at NOA and Tanagra. The analysis of the daily $T_{min}$ at the NOA and Tanagra stations during summer 2021 showed that the average UHI intensity was 4.9 ± 1.5 °C during the non-HW days. However, the average UHI intensity was 7.7 ± 0.9 °C (peaking up to 9 °C) during HW2021, suggesting an amplification of the UHI intensity by 2–3 °C during the episode. These results confirm the findings of previous studies, reporting positive synergies between UHIs and HWs, which result in the amplification of UHI magnitude under exceptionally hot conditions, especially during nighttime [45,56,60].

### 3.3. Ranking HWs at NOA (1900–2021)

In our effort to evaluate the severity of HW2021 on a centennial time scale, all identified HWs since 1900 (67 in total) were rated with respect to their scores in the indices defined in Section 2.2, namely their amplitude (HWA), intensity (HWI), and cumulative heat ($Heat_{cum}$) (Figure 4a–c). The HWs were also rated according to their duration (number of HW days) and the number of hours across each HW corresponding to 'at least very strong heat stress conditions', namely hours of the UTCI > 38 °C, according to the scale in Table 1 (Figure 4d). The figures are organized in descending order of $HWA_{max}$ (Figure 4a), $HWI_{max}$ (Figure 4b), $Heat_{cum,max}$ (Figure 4c), and hours of UTCI > 38 °C (Figure 4d). Figure 4a–d include only the HWs with scores > 50th percentile of each index, namely the first 33 ranked HWs. The HWs are distinguished according to the year of their occurrence. In the case of more than one HW in a, the HWs are further distinguished with the symbol 'a', 'b', 'c', and so on (as shown on the horizontal axis of Figure 4), with 'a' denoting the chronologically first HW in the particular year. For instance, the HW 2021-c in Figure 4a–d refers to the third HW in summer 2021 (28 July to 6 August 2021), as described in 3.2, namely to HW2021.

This processing revealed that the HW of 1987 (20–27 July 1987), the two HWs of 2007 (2007-a and 2007-b during 24–28 June 2007 and 19–27 July 2007, respectively), and the third HW of 2021 (28/7–6 August 2021) ranked among the first five positions more frequently than any other HWs since 1900 (at least in seven out of the eleven examined indices in Figure 4a–d).

Table 3 enables a more direct comparison between the top four ranked HWs with respect to their values in each HW index. Along with the indices presented in Figure 4a–d, Table 3 includes some additional metrics concerning human thermal comfort based on the scale of the UTCI in Table 1. In addition to the total number of hours under at least strong/very strong or extreme heat stress conditions across each HW, the maximum number of consecutive hours under the aforementioned heat stress categories was estimated for each HW.

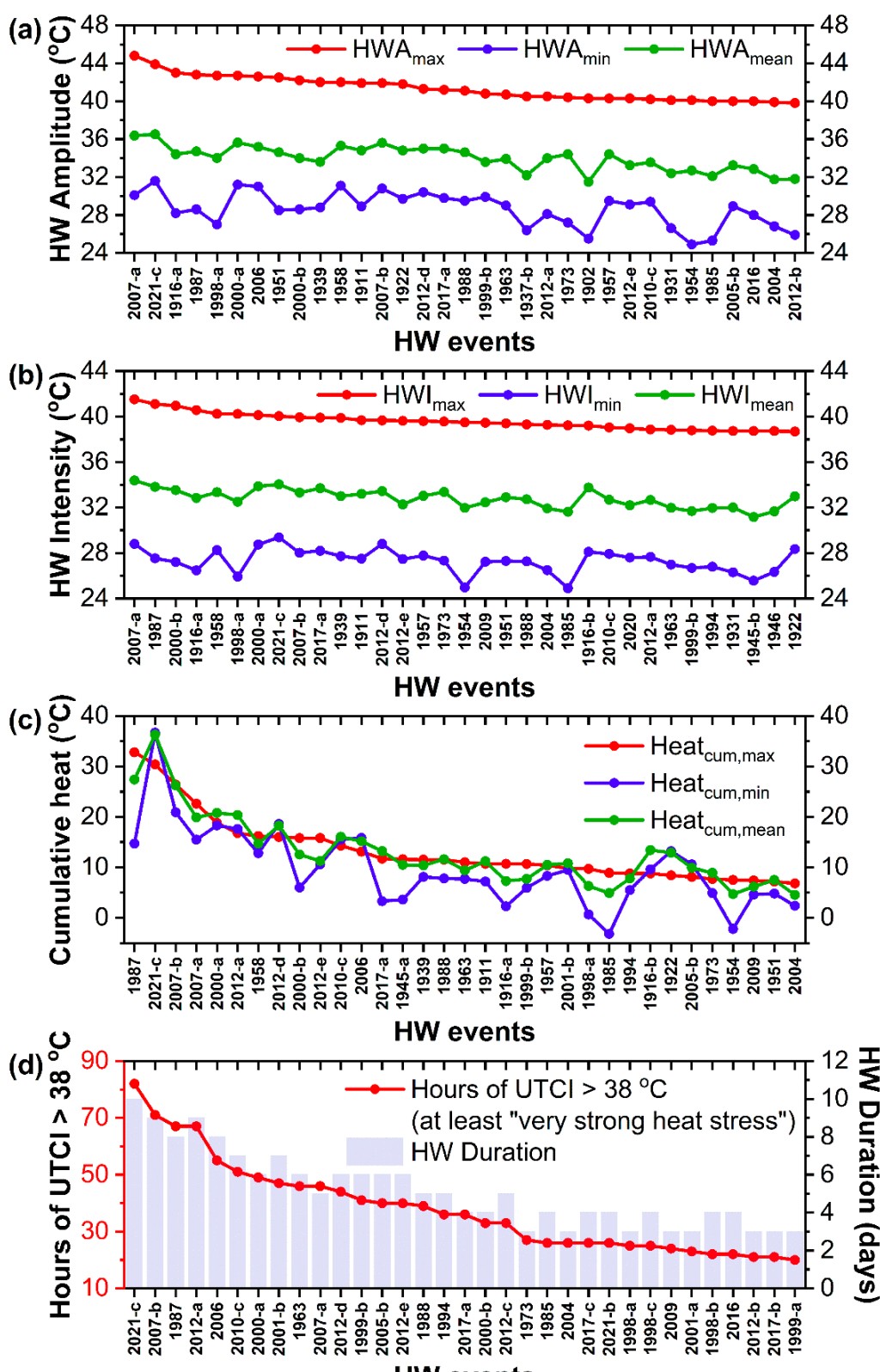

**Figure 4.** (**a**) Amplitude of HWs (HWA_max, HWA_min, and HWA_mean) in descending order of HWA_max; (**b**) intensity of HWs (HWI_max, HWI_min, and HWI_mean) in descending order of HWI_max; (**c**) cumulative heat of HWs (Heat_cum,max, Heat_cum,min, and Heat_cum,mean) in descending order of Heat_cum,max; (**d**) frequency (total hours) under 'at least very strong heat stress' during HWs and duration of HWs in descending order of heat stress frequency. (The graphs include the first 33 ranked HWs (>50th percentile) of HWA_max, HWI_max, Heat_cum,max, and heat stress frequency).

**Table 3.** Values of different HW indices and metrics for the four top-ranked HWs of 1987, 2007 (a and b), and HW2021. Bold fonts indicate the first-ranked value in each index.

| | HW1987 (20–27/7) | HW2007-a (24–28/6) | HW2007-b (19–27/7) | HW2021 (28/7–6/8) |
|---|---|---|---|---|
| Duration (days) | 8 | 5 | 9 | **10** |
| $HWA_{max}$ (°C) | 42.8 | **44.8** | 41.9 | 43.9 |
| $HWA_{min}$ (°C) | 28.6 | 30.1 | 30.8 | **31.6** |
| $HWA_{mean}$ (°C) | 34.7 | 36.4 | 35.6 | **36.5** |
| $HWI_{max}$ (°C) | 41.1 | **41.5** | 39.9 | 40.0 |
| $HWI_{min}$ (°C) | 27.5 | 28.8 | 28.0 | 29.4 |
| $HWI_{mean}$ (°C) | 33.8 | **34.4** | 33.3 | 34.0 |
| $Heat_{cum,max}$ (°C) | **32.8** | 22.6 | 26.4 | 30.4 |
| $Heat_{cum,min}$ (°C) | 12.3 | 14.0 | 18.2 | **33.7** |
| $Heat_{cum,mean}$ (°C) | 30.6 | 21.9 | 29.9 | **40.3** |
| Number of hours under at least strong heat stress (UTCI > 32 °C) | 101 | 63 | 104 | **122** |
| Number of hours under at least very strong heat tress (UTCI > 38 °C) | 70 | 46 | 71 | **82** |
| Number of hours under extreme heat stress (UTCI > 46 °C) | 3 | **8** | 1 | 7 |
| Maximum number of consecutive hours under at least strong heat stress (UTCI > 32 °C) | 14 | 14 | 13 | **16** |
| Maximum number of consecutive hours under at least very strong heat stress (UTCI > 38 °C) | 10 | 10 | 10 | 10 |
| Maximum number of consecutive hours under extreme heat stress (UTCI > 46 °C) | 2 | 4 | 1 | **6** |

In 11 of the 16 different metrics in Table 3, HW2021 was ranked first among all HWs since 1900, as a result of the combined effect of its intensity and duration. With a total duration of 10 days, HW2021 was the longest HW on record, followed by HW2007-b and HW2012-a, lasting 9 days (Figure 4d and Table 3). The highest $T_{max}$ during HW2021 (43.9 °C) did not surpass the previous long-lasting record value of 44.8 °C, observed during the early HW of 2007 (2007-a) [15]. Nevertheless, the nighttime temperature ($T_{min}$) of 31.6 °C and the daily average temperature ($T_{mean}$) of 36.5 °C recorded during HW2021 set new all-time records at NOA (Figure 4a, Table 3). Particularly high $T_{min}$ and $T_{mean}$ values were observed across the whole event, resulting in record-breaking values of the average intensity ($HWI_{min}$ and $HWI_{mean}$) and cumulative heat ($Heat_{cum,min}$ and $Heat_{cum,mean}$), as well (Figure 4c and Table 3). High intensities combined with the long duration of HW2021 contributed to very large values of $Heat_{cum,min}$ and $Heat_{cum,mean}$, which were much higher compared to any other HW on record (Figure 4c, Table 3). In particular, the nighttime cumulative heat ($Heat_{cum,min}$) added up to 33.7 °C during HW2021, being 2–3 times higher compared to previous top-ranked HWs (Table 3). Interestingly, the deadly HW of July 1987 (associated with at least 1000 excess deaths in Athens, [37]) ranked first only in cumulative heat based on the $T_{max}$ ($Heat_{cum,max}$ = 32.8 °C, Table 3), due to the unprecedented sequence of 8 consecutive days above 40 °C across the event. The high intensity and persistence of HW2021 dramatically increased the exposure of the human population to heat-related stress. HW2021 ranked first among all previous HWs with respect to the frequency (total hours of exposure) of at least strong and very strong heat stress (Figure 4d, Table 3). Moreover, for the first time, conditions of at least strong heat stress prevailed for 16 consecutive hours, while extreme heat stress conditions prevailed for 6 consecutive hours on 3 August 2021.

## 4. Discussion and Conclusions

Recent extreme heat waves and associated catastrophic impacts have greatly increased scientific interest on the topic, with relative research studies almost doubling within last

5 years [61]. Summer 2021 was undeniably an exceptionally extreme summer, with astounding heat shattering all-time records in different regions of the globe, including high latitude countries not acclimatized to heat [62]. The HWs of 2021 confirm early 2000s studies and model simulations projecting an increased intensity and duration of HWs [11,12]. According to World Weather Attribution, such extreme events would likely be impossible without human-induced climate change [63]. Recent research studies suggest that heat events exceeding previous records by large margins are very likely to occur in the coming decades [6]. Keeping global warming to below 2 °C above preindustrial levels is crucial for the protection of the global population from extreme heat. Dosio et al. [4] found that about 420 million more people would be exposed to severe heat waves under 2 °C compared to 1.5 °C of global warming.

A strong and persistent HW hit the Mediterranean from late July to early August 2021, affecting eastern Mediterranean countries such as Greece. The study evaluated the severity and special characteristics of this extreme event (HW2021) in Greece on a centennial time scale through the analysis of the historical climate records of the National Observatory of Athens, as well as records of national networks [57–59] at different Greek sites. Greece has witnessed severe heat waves in the past, even in the first half of the 20th century [38,40]. However, the severe heat waves in the last few decades have been more frequent and intense, following regional warming rates [40,41]. The current study revealed a five-fold increase in the number of HWs and HW days after 1990 at NOA, compared to their average values over the past period, 1900–1990.

A number of indices were selected to designate the severity of HWs and rank all identified HWs from 1900 to 2021, according to their scores in each index. In addition to broadly used indices concerning the duration and intensity of HWs, the study used the cumulative heat index to quantify and feature the excess heat experienced during the event. Note that excess heat does not refer to accumulated heat above normal (climatic) values across the HW, but above the upper 95th percentiles of the summer $T_{max}$ temperature distribution, already representing extreme heat values. The study adopted a fixed percentile value (37 °C) as a suitable threshold to identify a HW and estimate cumulative heat. The adoption of a different threshold (e.g., based on moving percentiles) would likely result in different HW identification and different estimates of cumulative heat. Nevertheless, the adopted threshold was selected to account for HWs inducing heat stress conditions, as was also the case for the choice of bioclimatic indices.

HW2021 rates as one of the most severe HWs in modern Greek history (at least since the initiation of systematic meteorological observations in the country), with daily maximum temperatures exceeding 46 °C at continental stations, surpassing long-standing records on a national scale. According to NOA observations and a number of adopted HW metrics, HW2021 ranks first among all identified heat waves in Athens since 1900 with respect to the duration of the event (10 days), highest nighttime ($T_{min}$) and daily average ($T_{mean}$) temperatures (reaching up to 31.6 °C and 36.5 °C, respectively), and accumulated heat, which exacerbated heat-related stress. Based on the UTCI, HW2021 ranks first on record with respect to the total hours (as well as consecutive hours) under strong/very strong and extreme heat stress. On 3 August, (the hottest day of the episode), the population was exposed to strong/very strong and extreme heat stress conditions for 16/10 and 6 consecutive hours, respectively. Other historic HWs in the country (in 1987 and 2007) scored high in indices related to daily maximum temperature ($T_{max}$). Accumulated excess heat, determined by both the intensity and duration of a HW, may have the most disastrous impacts on human health and ecosystems [7]. Nighttime temperatures during the event were much higher in Athens compared to non-urban stations, due to the additive effect of the urban heat island phenomenon, in contrast to daytime temperatures, which were higher at several continental sites of the country (Table 2). This confirms previous findings suggesting strong synergies between UHIs and severe HWs, evident in nighttime rather than daytime temperatures [45,56,60]. During severe HWs, large-scale synoptic processes and strong anomalies in the upper and lower atmosphere overwhelm mesoscale

or local effects, such as UHIs, during the daytime, in contrast to nighttime, when built environments release the absorbed heat. Thanks to the preparedness of authorities and agencies, increased public awareness, and local social acclimatization, HW2021 was not associated with numerous, heat-related excess deaths, as was, for instance, the case of the HW of July 1987 [37]. However, extreme heat in summer 2021 was linked with the occurrence of widespread, destructive wildfires in the country.

In addition to consecutive HWs, dry conditions prevailed during spring and summer 2021. According to NOA records, total precipitation in Athens from March to July 2021 was less than 30 mm, being lower by approximately 75% compared to the climatic value over this period. Prolonged hot conditions, in combination with dehydrated soil and the prevalence of strong winds, triggered the ignition and spread of catastrophic wildfires across the country, with dramatic environmental impacts and economic loss. More than 80 large forest fires were registered, which burned, in total, more than 1,300,000 acres of Greek forests and destroyed thousands of properties, forcing residents to evacuate. The fires destroyed peri-urban forests to the north of Athens and the largest part of northern Evia (the second largest island of the country). These were the worst wildfires in the country after the catastrophic fires of 2007 (also coming after consecutive HWs and drought conditions) that destroyed 2,800,000 acres, corresponding approximately to 2% of the Greek land surface [15]. Thus, HW2021 rates as one of the most severe HWs in modern Greek history, from the perspective of catastrophic impacts, as well.

The Mediterranean is responding quickly to climate change [18,29,31], making the area highly vulnerable to environmental risks, such as droughts and thermal risk, which are also anticipated to increase in the future [5,28–31]. Projected changes and environmental and health risks in the Mediterranean region were especially highlighted in the last report of the Intergovernmental Panel on Climate Change (IPCC), released on 28 February 2022 [64]. Recent research studies point towards the greater exposure of the population to heat-related risk in the southern European Mediterranean region [65]. Moreover, urban areas of the eastern Mediterranean, such as Athens, belong to the top-ranked European cities with regards to future heat-related risk [27,32].

Extreme HWs like HW2021 will very likely be more frequent in the area in the coming decades, threatening humans and ecosystems [3–5,15]. Fire risk may be exacerbated from combined projected droughts and hot conditions, along with the exposure of humans to heat-related stress [7,28–31,65]. The high, record-breaking nighttime and all-day temperatures in Athens during HW2021 underline the increased vulnerability of urban populations to thermal risk [66]. UHIs are closely linked to increased mortality during severe heat waves [37,67,68].

HWs, such as HW2021, pose new challenges and priorities for the protection of local societies and the environment from the adverse impacts of climate change. The detailed mapping of vulnerability in several sectors, preparation of strategy plans for the mitigation of risk, and adaptation is of paramount importance in Mediterranean countries, and especially highly urbanized areas such as Athens.

**Author Contributions:** Conceptualization, D.F.; methodology, D.F. and G.K.; software, G.K. and F.P.; formal analysis, D.F. and G.K.; investigation, D.F. and G.K.; data curation, G.K. and F.P.; writing—original draft preparation, D.F. and G.K.; writing—review and editing, D.F. and N.M.; visualization, G.K.; supervision, D.F. and N.M. All authors have read and agreed to the published version of the manuscript.

**Funding:** This research received no external funding.

**Institutional Review Board Statement:** Not applicable.

**Informed Consent Statement:** Not applicable.

**Data Availability Statement:** Climatic data at NOA are available by authors upon request. To access data for other Greek stations, visit https://www.meteo.gr or https://www.hnms.gr (accessed on 6 April 2022).

**Acknowledgments:** This work was supported by the action titled "National Network on Climate Change and its Impacts (CLIMPACT)", which is implemented under subproject 3 of the project "Infrastructure of national research networks in the fields of Precision Medicine, Quantum Technology and Climate Change", funded by the Public Investment Program of Greece, General Secretary of Research and Technology/Ministry of Development and Investments. The authors are grateful to all observers and operators of NOA for the maintenance of reliable, long climate records and to the *meteo* group of NOA for the free access of climatic data from NOANN. The authors are also grateful to the Hellenic National Meteorological Service for the provision of air temperature data.

**Conflicts of Interest:** The authors declare no conflict of interest.

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
