# Peer review of "The Extreme Heat Wave of Summer 2021 in Athens (Greece): Cumulative Heat and Exposure to Heat Stress"

_sustainability, doi:10.3390/su14137766_

Round 1
Reviewer 1 Report
The manuscript “The extreme heat wave of summer 2021 in Greece: cumulative heat and exposure to heat stress” submitted by Founda et al. is a documentary study that examines the rarity, amplitude and other special characteristics of the extreme heat wave event of summer 2021 in Greece by using a long station dataset in Athens. They put the event in a climatological context and compared this event with other historical events that hit in Greece. They found a significant upward trend for the occurrences of heat wave events since the 1990s, and in this background, the 2021 event was ranked top in terms of many key heat wave metrics such as the persistence and cumulative heat, etc. The findings of this papers provide key information regarding the heat wave events in Greece, which is valuable for relevant authorities and policy makers. The manuscript is written in a concise and easy-to-read manner. To me, this manuscript is suited for publication in Sustainability.
A major concern regarding the findings is the representativeness of heat event in Greece by using one meteorological station in Athens. The reviewer understands the long climatic record of the National Observatory of Athens (NOA) makes it possible to investigate the extreme event in a climatological context of more than 100 years. However, since the NOA is located in the urban area and greatly influenced by the urban island effect, the temperature, especially for the minimum temperature, is in sharp contrast with that in other stations (Table 2). A possible outcome could be an exaggeration of the upward trend of Tmin in recent decades shown in Figure 1. Therefore, using the records based only on one station to represent the Greece may unavoidably add uncertainties to their findings.
Another concern is the authors used a fixed threshold to define the heat wave. To me, this definition will be inevitably influenced by the annual cycle and exaggerate a heat event occurred in the peak of the annual cycle. An associated comment is the definition of “anomaly” in the manuscript, a similar question remains if the authors did not consider the annual cycle and used a fixed threshold to define the climatology.
Finally, I recommend the authors to start their main text with a figure showing the spatial distribution of this heat wave event (e.g. the maximum temperature) and mark the relevant meteorological stations used in Table 2 in the map.
Reviewer 2 Report
The manuscript tries to explore the 2021 summer heat wave in Athens, Greece. The manuscript is an extension of a prior published article by co-authors. I find the manuscript to be interesting and relevant to climate literature. Overall, the manuscript is well written, it is easy to read. The figures are well done and of good quality. I believe that the methodology needs to be carefully elaborated. For those reasons, it is recommended to be published in Sustainability after major revision.
There are some issues that can possibly be addressed to further improve the manuscript. Authors should go one step further in their analysis. The analysis of heat waves is only a case study for Athens. I believe that it would add value to the work if the heat waves' intensity were related to the heat island of Athens city. Here I suggest some points that the Authors may consider enhancing the manuscript:
· · The title is inappropriate, the text refers only to Athens and the wider region and not to the whole of Greece. Also, the analysis of the heatwave of summer 2021 is a small part of the analysis of the text. It is proposed to change the manuscript title.
· · Section 3.2 describes and analyses the heatwave of the 2021 summer in Athens. The text contains analysis and references, so I believe that it is not fit in section 2 "Materials and Methods". I think that if it is not a part of the authors’ analysis it should be moved to the introduction otherwise to the analysis of the results.
· · In Figure 1, the mean duration of heat waves (in days) should be added.
· · Line 193-201: The description is complicated and does not help in understanding the analysis. A diagram showing the frequency of heat waves would be helpful.
· In Figure 2, the anomaly axis should be symmetric. Moreover, the daily climatic means used for the estimation of the anomalies should be referred to in the manuscript.
· In Figure 3, the second axis of the line plots is inappropriate since it presents a similar magnitude to the first axis. It should be deleted. The second axis of Figure 2d can remain.
· In Table 3, the selection of the 4 top-ranked HWs are not obvious. According to Figure 3, 2012-a, 1916-a, and 1987 are also top-ranked HWs. The authors could justify their selection.
Reviewer 3 Report
It is a very well structured paper on heat waves in 2021 in Athens and Greece.
Authors presented heat waves in 2021 using several indices indicating the intensity, persistence and bioclimatic effects. They also compared the conditions during the strongest heat wave in 2021 in Athens and other places in Greece, pointing to the differences and explaining it by the effect of the urban heat island. This heat wave was also shown in long term perspective from the beginning of the 20th century. The data and methodology is well described, the arguments and the discussion of findings is coherent, balances and compelling. Because it is a description of one year heat waves, so the contribution to science is limited.
Round 2
Reviewer 1 Report
I appreciate the authors' efforts and they have adequately addressed my comments.
Reviewer 2 Report
The manuscript of the article entitled “The extreme heat wave of summer 2021 in Athens (Greece): cumulative heat and exposure to heat stress” has been revised according to the reviewers’ remarks. The title changed and the authors made an effort to reply point by point to reviewer’s comments. Their detailed reply managed to rebut any manuscript weaknesses.
Finally, I insist on changing figure 3 according to my initial comments, since the “flat” plot of the Tmax doesn’t change using the secondary axis. I believe the submitted version of the paper can be accepted and published in the SUSTAINABILITY after changing Figure 3.
